# The Role of Endometrial Stem/Progenitor Cells in Recurrent Reproductive Failure

**DOI:** 10.3390/jpm12050775

**Published:** 2022-05-11

**Authors:** Hannan Al-Lamee, Christopher J. Hill, Florence Turner, Thuan Phan, Andrew J. Drakeley, Dharani K. Hapangama, Nicola Tempest

**Affiliations:** 1Centre for Women’s Health Research, Department of Women’s and Children’s Health, Institute of Life Course and Medical Sciences, University of Liverpool, Member of Liverpool Health Partners, Liverpool L8 7SS, UK; hannan.al-lamee@liverpool.ac.uk (H.A.-L.); c.j.hill1@liverpool.ac.uk (C.J.H.); f.turner2@liverpool.ac.uk (F.T.); p.thuan@liverpool.ac.uk (T.P.); dharani@liverpool.ac.uk (D.K.H.); 2Hewitt Centre for Reproductive Medicine, Liverpool Women’s NHS Foundation Trust, Liverpool L8 7SS, UK; andrew.drakeley@lwh.nhs.uk; 3Liverpool Women’s NHS Foundation Trust, Member of Liverpool Health Partners, Liverpool L8 7SS, UK

**Keywords:** endometrium, stem/progenitor cells, adult stem cells, implantation, pregnancy loss, recurrent implantation failure, recurrent pregnancy loss, miscarriage, recurrent reproductive failure

## Abstract

Recurrent implantation failure (RIF) and recurrent pregnancy loss (RPL), collectively referred to as recurrent reproductive failure (RRF), are both challenging conditions with many unanswered questions relating to causes and management options. Both conditions are proposed to be related to an aberrant endometrial microenvironment, with different proposed aetiologies related to a restrictive or permissive endometrium for an invading embryo. The impressive regenerative capacity of the human endometrium has been well-established and has led to the isolation and characterisation of several subtypes of endometrial stem/progenitor cells (eSPCs). eSPCs are known to be involved in the pathogenesis of endometrium-related disorders (such as endometriosis) and have been proposed to be implicated in the pathogenesis of RRF. This review appraises the current knowledge of eSPCs, and their involvement in RRF, highlighting the considerable unknown aspects in this field, and providing avenues for future research to facilitate much-needed advances in the diagnosis and management of millions of women suffering with RRF.

## 1. Introduction

The endometrium is a highly dynamic, complex, and critical organ for successful human reproduction [1]. A multitude of harmonious processes allow for menstrual shedding, repair, and remodelling of the endometrium on a monthly basis, throughout a female’s reproductive lifespan, with receptivity of an embryo occurring for a finite time period in each cycle [1,2,3]. During this ‘window of implantation’ (WOI), the human endometrium, and viable blastocyst, must undergo an elaborate and tightly orchestrated crosstalk to achieve successful implantation, and for the endometrium to support an ongoing pregnancy [4].

Recurrent reproductive failure (RRF) encompasses two separate but related conditions, recurrent implantation failure (RIF), and recurrent pregnancy loss (RPL) [3,5,6,7]. RIF affects around 10% of women undergoing embryo transfers (ETs) during in vitro fertilisation (IVF) treatment [8], and the generally accepted definition is three or more failed ET attempts [9,10]. RPL has a prevalence of 1–2% [11,12], with around 50% of unexplained aetiology [13]. RPL has no universally accepted definition when referring to the number of pregnancy losses; however, the loss of either two [11,12] or three [14] consecutive pregnancies are most widely accepted. Both conditions pose major challenges for clinicians who are left with no clear definitive or effective management options, causing extreme distress for patients who may also have many unanswered questions [6,7,15]. The causes of RRF are multifactorial [16,17]; however, the proposed pathophysiological mechanisms vary according to both maternal age and gestational age when the pregnancy loss occurs [18]. Risk factors associated with both RIF and RPL include uterine abnormalities [19,20], tubal pathology [21], acquired and hereditary thrombophilia [22], endometriosis [23], suboptimal endometrial thickness [24,25], metabolic and autoimmune disorders [26,27,28], chronic infection and immunological and lifestyle factors, etc. [12,29,30,31,32,33] (Figure 1A). RIF is thought to be related to decreased endometrial receptivity [18,34,35] and RPL to an unselective or permissive endometrium with an impaired decidualisation, leading to disrupted embryo-endometrial dialogue, with a consequential lack of natural embryo selection and subsequent inappropriate implantation of a non-viable embryo [12,36,37]. Increasing evidence is now emerging that suggests a strong endometrial cause for RRF, with reports suggesting that two-thirds of RIF is secondary to inadequate endometrial receptivity [3,6,18,38], and many other studies also propose an association between unexplained RPL and a defective endometrium [3,6,17,32,39] (Figure 1B).

Endometrial glands are essential for the establishment of a pregnancy, with glandular topography and secretions integral to embryo attachment, and thus, are vital for the subsequent establishment of the decidua [40,41,42,43,44]. So far, studies of epithelial endometrial stem/progenitor cells (eSPCs) have been based on the long-accepted presumption that the human endometrial glandular architecture arise from single blunt-ended tubes within the basalis, similar to the architecture of the intestinal crypts [45,46,47,48,49]. Recent three-dimensional (3D) reconstruction of the endometrium has altered our understanding of the glandular microarchitecture, showing mycelium-like, horizontal branching networks within the deep basalis layer [48,50,51,52]. This newly discovered glandular architecture is important to consider in light of pathologies such as RRF. Endometrial glands and stroma are thought to arise and regenerate from SPC populations that are located within the relatively static basalis layer, which is undisturbed at menstrual shedding [50,53,54,55,56,57]. Aberrant eSPCs have been associated with endometrial proliferative disorders such as endometriosis and endometrial carcinoma [54,58,59,60,61]. It is possible that RRF may also arise from an abnormal glandular network, secondary to aberrant eSPCs. Dysfunctional endometrial repair may present in women with infertility, RIF, or RPL, and may be implicated in the aetiology of clinical conditions such as Asherman’s syndrome (AS) and a persistently thin endometrium [62]. AS is a condition whereby intrauterine adhesions result in a thin or defective endometrium, due to probable extensive destruction of the basalis glands (usually protected from external influence), thereby impacting the functionalis, and consequentially embryo implantation and pregnancy [48,63]. Evidence is emerging that RPL is associated with a depletion of highly proliferative mesenchymal cells that act as decidual precursors and are likely to originate from bone marrow-derived mesenchymal stem/stromal cells (BMDMSCs) [64]. The use of exogenous, autologous, or allogenic stem cells has been proposed in regenerative medicine, including the treatment of atrophic or scarred endometrium [62,65]. These have shown promise for the clinical application of SPC therapy in endometrium-related gynaecological pathologies and fertility, with reports of BMDMSCs used to improve endometrial thickness and vascularity, restore menses, and aid conception in patients with AS or endometrial atrophy [65,66,67,68,69].

Although evidence is now emerging on the possible link between non-eSPCs and RRF, little is known about the resident eSPCs and their association with these pathologies [62,70]. Evidently, much work is required in the characterisation and clinical application of eSPCs in such disorders. However, their key role in the regeneration of endometrial glands and luminal epithelium (LE) [50,71], and their role in stromal decidualisation [72,73], would suggest that they are likely to have an important function in endometrial receptivity, embryo implantation, and the support of an ongoing pregnancy.

## 2. Scope of This Review

This review sought to examine and summarise the current knowledge base for eSPCs and their potential involvement in RRF, to identify important voids in the literature, and to highlight avenues for future research.

Non-resident endometrial SPCs (including BMDMSCs) and non-endometrial MSCs (such as umbilical cord-derived, amniotic-derived, or adipose-derived MSCs) are other additional stem cell populations that have been previously explored in relation to endometrial regeneration [62], and lie beyond the scope of this review. Only those SPCs derived from the endometrial cell types (e.g., epithelial and stromal) will be included in this review with regards to the aetiology of RRF.

## 3. Materials and Methods

An extensive literature search was performed across multiple databases, including PubMed, Web of Science, EMBASE, and Scopus, and spanned publications from inception to February 2022. Studies were selected using keywords associated with eSPCs, RRF, RIF, and RPL. Additionally, all references cited within other relevant publications were screened. Publications related to all keywords were included, and this included both human and animal studies.

## 4. Endometrial Stem/Progenitor Cells

The remarkable regenerative capacity of the endometrium is widely believed to arise from SPCs residing within the deeper basalis layer, which remains intact following menstruation and the menopause [48,54,56,57,74,75,76,77,78]. The functionalis layer can re-grow from a thickness of 1–2 mm, following menstruation, to 14 mm during the secretory phase of the cycle [79]. Remarkable restoration is also seen following parturition, iatrogenic surgical destruction of the endometrium (such as endometrial ablation), and within the post-menopausal endometrium following exposure to oestrogen hormone replacement therapy [77,80,81,82]. eSPCs are thought to be the key players in driving this impressive proliferative capacity and cell turnover [48,54,56,57,83,84].

Since the existence of endometrial adult stem cells (ASCs) was first postulated [74,75], much work has been invested into identifying and characterising eSPCs in both human endometrium and in mouse models [50,53,54,71,85,86,87,88,89,90,91,92,93,94,95,96]. In 2004, rare populations of clonogenic epithelial and stromal cells were first reported [53]. Identifying and examining human eSPCs is challenging due to their scarcity within the tissue, their lack of specific markers allowing for their isolation, and their change in phenotype when taken out of their highly unique endometrial microenvironment [57,97]. Studies attempting to identify these cells have mainly explored their distinguishing functional properties, such as their clonogenic ability, self-renewal capacity, proliferation and differentiation capacity, label retention, and tissue reconstitution assays [50,54,56,71,87,88,89,90,98,99,100,101].

Work identifying endometrial stromal ASCs (also known as endometrial mesenchymal stem cells (eMSCs)) was first initiated by Gargett and colleagues, who examined cells displaying higher clonogenic colony-forming capacity in vitro [53]. Endometrial stromal ASCs are seen to show similar properties to BMDMSCs and are multipotent, displaying the capability to differentiate into fat, bone, cartilage, skeletal muscle, and smooth muscle [100,101,102,103]. Endometrial stromal ASCs have been well characterised, with CD146^+^ platelet-derived growth factor receptor beta (PDGFRβ^+^) [93,102] and sushi domain containing-2 (SUSD2^+^), as specific markers for their enrichment [96]. More recently, nucleoside triphosphate diphosphohydrolase 2 (NTPDase2) has been detected in just the perivascular SUSD2^+^ cells and not the rest of the stromal fraction, and thus has been proposed as a marker for eMSCs located in the endometrial basal layer [104]. SUSD2^+^ cells (also referred to as W5C5^+^) are found to increase during the proliferative phase, suggesting their involvement in regeneration of the functional stroma [96]. They are also seen within postmenopausal endometrium treated with oestrogen, and reconstitute endometrium under the kidney capsule of xenografted mice [96]. Their location is now widely accepted to be within the pericyte and perivascular cells of the endometrial basalis and functionalis, with transcriptomics and secretomics of SUSD2^+^ cells confirming their perivascular phenotype [56,93,96,104].

Unlike endometrial stromal ASCs, epithelial eSPCs are yet to be conclusively defined, with specific universal markers remaining elusive [57]. Work focused on characterising the epithelial eSPC population was first initiated within our laboratories, with the breakthrough discovery of cell surface marker stage-specific embryonic antigen-1 (SSEA-1) to demarcate the basalis epithelial cells [54]. The proposed SSEA-1^+^ epithelial eSPCs show higher telomerase activity and longer telomere lengths, a greater propensity to generate spheroids in 3D cell culture, and low expression of steroid hormone receptors [54,61]. Subsequent work has also suggested N-cadherin to be an epithelial eSPC marker. It enriches for clonogenic epithelial cells showing greater self-renewal capacity and quiescence, with the ability to generate large gland-like spheroids [105] (Figure 2). These characteristics correlate with a more primitive or SPC phenotype. With in vivo immunostaining of SSEA-1^+^ and N-cadherin^+^ endometrial epithelial cells in full thickness endometrium, Nguyen et al. have proposed that there is a spatial relationship between the two cell types, with SSEA-1^+^ cells located closer to the basalis–functionalis interface, whilst N-cadherin^+^ cells situated deeper in the basalis. With this observation, the authors proposed a possible differentiation hierarchy amongst the epithelial eSPCs [105]. Prior to these findings, Musashi-1 (an ASC marker) had also been immunolocalised to single epithelial cells within endometrial glands, and small clusters of stromal cells [94]. Musashi-1^+^ cells were found mainly within the basalis of the proliferative phase endometrium, supporting its possible SPC role; however, no functional data is available for these cells, thus the evidence for Musashi-1 as an eSPC marker is limited [94]. Nuclear WNT signalling pathway molecules, axis inhibition protein 2 (AXIN2) [106,107], SRY-box 9 (SOX9) [54], and nuclear ß-catenin [54] are also seen to be expressed by basalis glands; however, nuclear markers are not useful for the prospective isolation of epithelial eSPCs for important functional assessment, and therefore surface markers for their identification are required. More recently, leucine-rich repeat-containing G protein-coupled receptor 5 (LGR5) (a surface marker for intestinal epithelial stem cells) has been demonstrated to be expressed by a subset of epithelial cells within the LE, as well as in the glands of the basalis layer. However, due to the lack of reliable antibodies for immunology-based sorting, functional data for these cells are not yet available [71]. The proposed current understanding of eSPCs is summarised within the pictorial representation below (Figure 2).

eSPCs have been postulated to be involved in the pathogenesis of proliferative gynaecological disorders such as endometriosis [61,74,84,108,109]. Studies have demonstrated the expression of proposed epithelial stem cell markers SSEA-1 and nuclear SOX9, nuclear ß-catenin, and Musashi-1 in ectopic endometriotic lesions [54,61,94], which has shed light on the pathogenesis of the disease [61]. Endometriosis is associated with infertility [110], suggesting that the presence of atypical endometrial ASCs may be a possible common aetiology between these disorders, resulting in abnormal eutopic endometrium, as well as giving rise to the ectopic endometriotic lesions in endometriosis patients. In addition to the proliferative disorders of the endometrium, it is therefore likely that other disorders, associated with aberrant regeneration and remodelling of the functional layer of the endometrium, including infertility, RIF, and RPL [70], arise from abnormalities within the endometrial ASC population. Similar detailed examination of the involvement of eSPCs in RRF is expected to enhance our understanding of the pathogenesis of the disease.

## 5. Endometrial Stem/Progenitor Cells and Recurrent Reproductive Failure (RRF)

Two studies have identified the role of eSPCs in RRF (RIF and RPL), with current knowledge limited to eMSCs. To date, eMSCs have been suggested to play a role in endometrial receptivity and regeneration, both of which are essential to successful pregnancy.

### 5.1. Recurrent Implantation Failure (RIF)

Considering the well-established endometrial contribution to RIF, eSPC abnormalities are expected to be involved in generating a persistently abnormal functionalis layer. Unlike in RPL, the endometrium of women with RIF is proposed to be hostile and less receptive to embryo-attachment [18,34,35]. Studies have suggested that eMSCs express markers of endometrial receptivity, such as adhesion molecules. Endometrial stromal cells are known to express receptivity markers *HoxA11* and *Noggin* during the WOI [111]. In 2020, five markers of receptivity (*ITGβ1* [112], *RAC1* [113], *HOXA11* [111], *ITGβ3* [112,114], and *NOGGIN* [111]) were compared between menstrual blood samples taken from women with RIF, versus fertile controls, revealing different patterns of expression. This observation led the authors to propose that aberrant eMSCs contribute to altered endometrial receptivity in women with RIF [115]. eMSCs from menstrual blood were isolated and characterised based on their morphology and behaviour, expression of specific surface markers (CD44, CD31, CD34, CD73, CD90, and CD105), and capacity to differentiate, all of which was based on the International Society for Cellular Therapy statement [115,116]. Notably, significantly higher expression levels of both *HOXA11* and *ITG**β**3* were seen in the eMSCs of women with RIF [115]. *HOXA11* encodes a transcription factor and is involved in proliferation, differentiation, and embryologic development of the endometrium [117], therefore, *HOXA11* is thought to be important in establishing pregnancy [111,118]. *Rac1* expression is necessary for eMSC migration and is associated with increased cell motility at the site of implantation [113]. Integrins are adhesive molecules also highly expressed by MSCs and were found to play a role in embryo implantation and development [39,112,114,119]. Integrins play a vital role in extracellular matrix adhesion and therefore are vital in embryo implantation [112,120], with integrin β3 proposed as a useful tool to predict the success rate of assisted reproductive technology (ART) [112]. In agreement with this manuscript, recent work from another group also identified an increased expression of *HOXA11* antisense RNA in the endometrium of women with RIF, leading to downregulated decidualisation [121]. Contradictory to this, *Hoxa11* knockout mice are known to be associated with uterine factor infertility [117]. As for integrin β3, the available evidence is conflicting: higher expression has been shown in eMSC from RIF endometrial tissue samples [115], but, in a larger study, low expression in endometrial tissue has been associated with lower pregnancy rates [122]. It is apparent that *HOXA11* and integrin β3 contribute to uterine receptivity and are therefore relevant to RIF. *Hoxa11* is expressed in mouse uterine stromal cells during the receptive phase and during post-implantation decidualisation, suggesting a possible role in receptivity, implantation, and decidualisation [118]. *Hoxa11^-/-^* mice have hypoplastic uteri, fewer endometrial glands, and do not express *Lif*, which is essential for the progression of a normal pregnancy [111,123]. However, the mechanisms behind how specific eSPCs contribute to the observed abnormal endometrial *HOXA11* and integrin β3 levels, remains undefined. The results from Esmaeilzadeh et al. (the only study concentrating on eMSCs and RIF) should be interpreted with caution as the number of women who participated in the study was very small (n = 5) and the control group utilised (n = 3) was extremely heterogenous, almost overlapping with the RIF group (history of less than three miscarriages and at least one live birth after ART cycle(s) (two of the controls had a child after their second ART cycle and the other after their third cycle) [115]. In addition, the technique used to isolate and identify the eMSCs is questionable as they did not use the previously characterised eMSC markers (CD146^+^PDGFRβ^+^, SUSD2^+^, or NTPDase2), raising concerns regarding the appropriateness of the methodology and reliability of only eMSCs being isolated.

### 5.2. Recurrent Pregnancy Loss (RPL)

Similarly to RIF, aberrant endometrium is expected to contribute to the pathology of RPL. RPL is proposed to be associated with a hyper-receptive endometrium, permissive to even a sub-optimal blastocyst [124], with possible eSPC aberrations in the LE and functionalis, which are the first layers of contact for an incoming embryo [48]. To explore the eMSC population in women with RPL, Lucas et al. (2016) isolated perivascular and non-perivascular human endometrial stromal cells (W5C5^+^ and W5C5^−^, respectively) from luteinising hormone (LH) timed mid-luteal phase endometrial biopsies [125]. W5C5 was identified in 2012 as a single marker capable of purifying eMSCs possessing MSC properties and reconstituting endometrial stromal tissues in vivo [96]. No differences in the number of W5C5^+^ perivascular cells between the women with RPL and control samples were noted. However, there was a significant decrease in the clonogenicity of W5C5^+^ and W5C5^−^ cell populations from RPL samples when compared to fertile controls. No clonogenic W5C5^−^ cells were recovered from 42% of the RPL sample versus 11% of the control samples. In the cohort studied, the presence of clonogenic W5C5^+^ or W5C5^−^ cells were adversely associated with the number of previous miscarriages [125]. This means that controlling an eMSC deficiency could be considered a possible approach to prevent RPL. However, the contribution of the epithelial eSPCs to RPL has not yet been examined.

The discovery that a deficiency of eMSCs is connected to RPL has revealed new insights into pregnancy failure mechanisms. This has posed new questions concerning the pathways that control the maintenance of eMSCs from one pregnancy to another pregnancy.

## 6. Applications in Clinical Management of RRF

eSPCs have been found to play a crucial role in the regeneration and repair of the endometrium [54,126,127]. We therefore would envisage eSPCs to be involved in the persistent aberrations in the functionalis layer of each successive cycle, associated with the implantation process and subsequent pregnancy establishment in RRF. However, the available evidence for detailed knowledge from basic science studies on functional or phenotypic abnormalities of these specialised cells is still lacking to guide their safe clinical translation. It is therefore unsurprising that trials of eSPC therapies for women with RRF are currently not available. Therefore, studies are urgently required to explore the fundamental role of eSPCs in the pathogenesis of RIF and RPL, as well as to assess their therapeutic potential in the management of these conditions. Furthermore, common and challenging clinical conditions that are related to RRF include thin endometrium and AS, for which studies have been undertaken to aid clinical management with the use of eSPCs [62,126].

Persistently thin endometrium is a major challenge within the fertility setting [128,129,130,131], exemplified by the findings that, with each millimetre of decrease in endometrial thickness under 8 mm, a significant decrease in clinical pregnancy and live birth rates and dramatically increased pregnancy loss rates after achieving a clinical pregnancy occur [132,133,134]. The causes of persistently thin endometrium are unknown, with no studies to report if the deficiency is of the functionalis or basalis, epithelial eSPC or eMSC. However, the malfunctioning of eSPCs is postulated to have a role [135], and many studies exist targeting SPCs as a possible treatment avenue [136,137,138,139,140]. Exogenous oestradiol supplementation and various adjuvant therapies have been proposed to promote endometrial regeneration (such as aspirin, sildenafil, tamoxifen, vitamin E, pentoxifylline, L-arginine, or platelet-rich plasma) presumably by influencing eSPC activity; however, they all lack robust evidence and definitive efficacy for their proposed use [24,141]. A thin endometrium is known to be associated with RRF [24] and it may be idiopathic in nature [142,143] or possibly associated with inflammatory/iatrogenic causes such as intrauterine adhesions after a surgical procedure [24]. Post-surgical thin endometrium may be due to the removal of the stem cell rich basalis during extensive curettage, or following ablation [135]. The use of eSPCs to enhance regeneration of the thin endometrium can be expected to provide a novel solution to the ongoing dilemma of ‘thin endometrium’ by encouraging more efficient regrowth. We postulate that the eSPC would provide a stock of eSPCs able to integrate into the endometrium, thus increasing the resident eSPC pool. Alternatively, it may provide the appropriate niche for the existing eSPCs to function more efficiently, with the desired end point being the availability of the appropriate bulk of functional eSPCs, and provision of a sufficient amount of their progeny, on a monthly basis. In 2019, Hu et al. investigated the effect of menstrual blood-derived stem cells (MenSCs) on endometrial repair following mechanical injury in mice [144]. They found that mice treated with MenSC exhibited significantly higher expression of endometrial keratin, vimentin, and vascular endothelial growth factor, with an increased endometrial thickness, and increased pregnancy rates [144]. This was potentially due to the MenSC providing a better scaffold for cells with improved blood supply, cellular integrity, and resistance to cell damage. This evidence supports the notion that MenSC therapy improves the eSPC niche. Zhao et al. and Jing et al. also demonstrated that injecting BMDMSCs into the endometrial cavities of rats, resulted in increased endometrial thickness [137,145]. Injecting primitive MenSC or BMDSC cells seem to be beneficial with an intriguing mechanism of action that is yet to be clarified. We could hypothesise that the injected cells cause activation of quiescent resident eSPCs, enhancing the important support from the stem cell niche, or they may promote recruitment of new SPCs to the endometrium, rectifying a postulated deficient eSPC pool.

Currently, treatment for AS is usually surgical, with hysteroscopic adhesiolysis [146,147]. In severe cases, however, surgical treatments often fail, requiring multiple repeat hysteroscopic procedures, a 29% risk of recurrence, and a low post-treatment pregnancy rate of a mere 25% in these severely affected cases [148,149]. In 2021, the phenotypic and biological characteristics of eMSCs (co-expression of CD140b and CD146) from healthy fertile women, and women with intrauterine adhesions was compared [150]. The number of eMSCs in women with intrauterine adhesions was reported to be significantly lower compared with healthy fertile women [150]. Furthermore, the colony-forming capacity, migration, invasion, angiogenic/capillary formation, and immunosuppressive abilities of the eMSCs was also significantly lower, possibly affecting their ability to migrate and regenerate at the site of injury [150]. No evidence is available regarding the loss of eSPC stem cell niche being a reason for the deficient endometrium, and additional work needs to be done on this plausible theory. Although Min et al.’s study showed a significant decrease in the number and biological function of eMSCs in women with intrauterine adhesions, whether these disorders can be treated by eMSC supplementation requires further validation. Due to the main features of AS being the lack of basalis stem cell niche and fibrotic adhesions between the opposing uterine myometrial surfaces, the number of cells available to be harvested from AS women are likely to be low. Since the functional assays used in the experiments presented in the above study are directly relevant to the number of cells that can be harvested and used in in vitro assays, the expected reduction in number of cells harvested from AS endometrium could directly affect the assay data. In 2016, Zhang et al. applied human eMSCs derived from menstrual blood into mouse models. The authors concluded that stem cells derived from menstrual blood may have a role in repairing the damaged endometrium, which could be due to the engagement of angiogenesis mediated by stem cells derived from menstrual blood [151]. Therapy with autologous eMSCs derived from menstrual blood has also shown promise, accelerating the healing process of damaged endometrium by inducing self-renewal, differentiation, angiogenesis, anti-inflammation, and immunomodulation [127]. Numerous issues must, however, be addressed before treatments can be introduced to clinical practice, such as transplanted cell dosage, and administration route.

RPL has been associated with the loss of eMSCs [125]. In 2020, Tewary et al. conducted a randomised, double-blind, placebo-controlled feasibility trial, administering either sitagliptin or placebo to women with RPL, proposing to target BMDMSCs and eMSCs [152]. Sitagliptin is a dipeptidyl peptidase 4 (DPP4) inhibitor, which may encourage the migration and engraftment of BMDMSCs to the site of tissue injury. DPP4 inhibitors are commonly used oral hypoglycaemic medications, which work by blocking the action of DDP4, thereby increasing concentrations of active incretin hormones, and improving glycaemic control [153,154]. They have been seen to promote tissue regeneration following injury [152,155]. Their aim was to determine the impact of sitagliptin on the abundance of eMSCs and on endometrial decidualisation in women with RPL. By using colony forming unit assays, the authors concluded that sitagliptin increases eMSCs and decreases decidual senescence in women with RPL during the mid-luteal phase [152]. Decidual cells, but not senescent decidual cells, are required for a continuing pregnancy, and are dependent on continuous progesterone signalling, therefore, the decreased decidual senescence would lead to increased promotion of decidualisation by progesterone, successful implantation, and potentially an increased ability to maintain a pregnancy. Although further work is required, these findings have shown promise that eSPCs could be a potential target for treatment in women with RRF. The available evidence that endometrial plasticity can be pharmacologically enhanced through improving the number of eMSCs is encouraging.

## 7. Avenues for Future Research

Although we strongly believe that eSPCs have a critical role to play in the pathology of RRF, evidence for direct eSPCs involvement in RRF is currently lacking. Available data is limited, sometimes contradictory, and of varying quality. Furthermore, the lack of clarity on the precise phenotypical and functional properties, origin, and location of eSPCs contributing to normal endometrial regeneration, hinders progress in the identification of eSPC aberration, specific to RRF, and thus, impedes the advancement of potential eSPC-based therapeutic avenues.

The importance of understanding the critical role of the endometrium in the implantation and success of ongoing pregnancy is acutely apparent by the knowledge that only 50% of euploid embryos accomplish implantation [156]. In addition, 1–2% of women who do conceive (either spontaneously or using ART methods) are found to experience an unexplained loss of a karyotypically normal early pregnancy [12].

Currently, the majority of research exploring stem cells in unexplained infertility, RRF, or endometrial regenerative medicine is mainly restricted to non-endometrial stem cells [62]. This primarily includes MSCs derived from the bone marrow [67,68,157,158], umbilical cord [140,159,160,161], placenta [162,163,164,165,166], or adipose tissue [167,168,169,170]. Very limited data exists on eMSCs and their relationship to RRF, with no current evidence for the role of human epithelial eSPCs in this devastating condition. Whilst it is important to understand the role of all types of stem cells within the endometrium of women with RRF, it is essential to understand the role of resident eSPCs from which the two primary endometrial cell types arise [100]. Further work is required to investigate the role of human stromal and epithelial eSPCs in both RIF and RPL, to aid in diagnosis and provide translatable treatment options.

The LE (the endometrial layer that communicates first with the incoming blastocyst) and endometrial glands (which provide an abundance of secretions to enable successful implantation) have been proven to be essential for pregnancy establishment [40,41]. The LE and endometrial glands are both thought to originate from epithelial eSPCs within the basalis layer [50,54,57,71]. More work is evidently required to identify and characterise the epithelial eSPC population within the normal endometrium of fertile women, which will allow, in the future, further study on their, possibly, aberrant role in women with RIF and RPL. If/when robust epithelial eSPC surface markers can be identified, a possibility to selectively isolate these eSPCs, and use them in diagnostic and therapeutic technologies, would become apparent. Our new understanding of the endometrial glandular microarchitecture [48,50,51,52] supports the theory that primitive, quiescent eSPCs reside in the mycelium-like basalis glands and give rise to the functional glands that grow vertically from these branches. Characterisation and localisation of epithelial eSPCs should be re-considered and re-examined in light of this new discovery. It may be that patients who experience RIF and RPL have an aberrant or underdeveloped horizontal network of basalis glands, and therefore an abnormal eSPC population, leading to defective regeneration of the functional endometrium and LE [48,51,52].

There are many technical and ethical challenges to overcome in defining the intricacies of human embryo implantation, and there are still many aspects that are not fully understood. Much of our knowledge of embryo implantation has stemmed from animal studies and in vitro two-dimensional (2D) monolayer culture systems, neither of which accurately represent the in vivo endometrial or embryo physiology, or environment [48,171]. 3D investigation of the human endometrium and embryo implantation is an important avenue for future research [48,51,52]. The advent of ‘organoids’ has led to huge advances in in vitro 3D modelling techniques [171,172,173]. Endometrial organoids arise from single or small clusters of cells and resemble endometrial tissue ex vivo [174]. They are grown in a supportive hydrogel droplet (such as Matrigel), which acts to mimic the extracellular matrix in vivo, alongside complex growth media designed to recapitulate the endometrial niche [172,174]. This 3D culture system has been successfully adapted to derive embryo-like organoids (known as blastoids) [175,176,177], trophoblast organoids modelling placental development [178,179], and gland-like organoids originating from endometrial epithelial cells [54,172,180,181,182,183]. Future research can utilise these 3D modelling systems in order to study endometrial glandular development and embryo implantation in patients with RIF and RPL.

## 8. Discussion

RIF and RPL are devastating and challenging conditions, with many unanswered questions remaining. Evidence is now starting to emerge that eSPCs may be implicated in the aetiology of these conditions, giving rise to an abnormal endometrium, and therefore a suboptimal environment to support an ongoing pregnancy. Advances in endometrial stem cell biology and novel seminal discoveries, such as the 3D histoarchitecture of human endometrial glands [50,51], have allowed us to start to identify and isolate different eSPC populations. More work is, however, required in order to fully characterise them in vivo and establish the stem cell hierarchy. In particular, new highlights on the 3D microarchitecture of the human endometrium have provided novel insights into the accurate structure and arrangement of the endometrial glands [48,50,51,52]. This may provide new insight and explanations for the location, mechanisms and possible aberrations of endometrium-associated conditions, such as RRF.

## Figures and Tables

**Figure 1 jpm-12-00775-f001:**
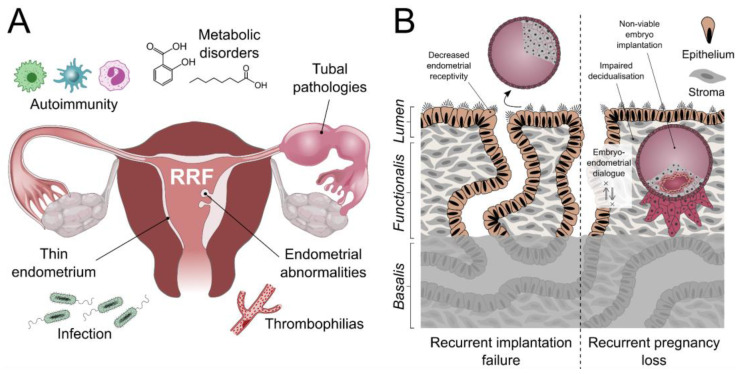
Aetiology of recurrent reproductive failure. (**A**) Risk factors for recurrent reproductive failure (RRF). (**B**) Demonstrable causes of recurrent implantation failure and recurrent pregnancy loss. The contribution of basalis-resident stem/progenitor cells and glandular architecture to RRF pathophysiology has yet to be fully explored.

**Figure 2 jpm-12-00775-f002:**
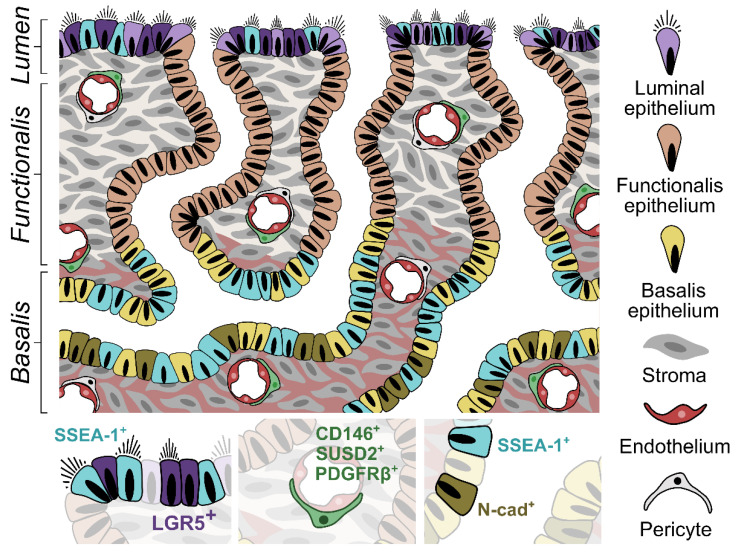
Proposed stem cell niches of the human endometrium. Stromal stem cells reside throughout the superficial and deep endometrial tissue as CD146^+^, sushi domain containing-2 (SUSD2^+^), and platelet-derived growth factor receptor β positive (PDGFRβ^+^) perivascular cells. Putative epithelial stem cell populations have been described in the basalis layer, a region characterised by expression of progenitor markers, including leucine-rich repeat-containing G protein-coupled receptor 5 (LGR5), SRY-box 9 (SOX9), axis inhibition protein 2 (AXIN2), and nuclear β-catenin. Specific cells within the deeper basalis glands express N-cadherin, whilst stage-specific embryonic antigen-1 (SSEA-1) is characteristic of more superficial basalis regions.

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
