# Peer review of "The Role of Endometrial Stem/Progenitor Cells in Recurrent Reproductive Failure"

_jpm, 2022, doi:10.3390/jpm12050775_

Round 1

Reviewer 1 Report

This is a through, well-written, and well-cited review on the contributions of SPCs to recurrent reproductive failure. I have very few critiques or items to add to this manuscript, with the exception of adding a few references to support certain of the Authors’ claims, lest they resemble setting up a straw-man argument.

  • Lines 70-72: cite references
  • Lines 94-95: cite references
  • Line 157- typo with “remining”
  • Lines 157-165: reference figure 2 here instead of just at the end of the paragraph, as I was about to propose a diagram, but continued to scroll.
  • Line 332-334: consider moving the description of AS to where it is first introduced in the manuscript
  • Line 368: define DPP4

Author Response

Reviewer 1

This is a through, well-written, and well-cited review on the contributions of SPCs to recurrent reproductive failure. I have very few critiques or items to add to this manuscript, with the exception of adding a few references to support certain of the Authors’ claims, lest they resemble setting up a straw-man argument.

We thank the reviewer for their helpful comments.

  • Lines 70-72: cite references

Thank you, we have now cited appropriate references (see line 73).

  • Lines 94-95: cite references

Thank you, we have now cited references for these lines (see line 99).

  • Line 157- typo with “remining”

Thank you for this observation, we have changed this accordingly to ‘remaining’ (see line 160).

  • Lines 157-165: reference figure 2 here instead of just at the end of the paragraph, as I was about to propose a diagram, but continued to scroll.

Thank you for this helpful comment, we have now referenced figure 2 on line 168.

  • Line 332-334: consider moving the description of AS to where it is first introduced in the manuscript

We thank the reviewer for this helpful suggestion, we have now moved the description to lines 85-88.

  • Line 368: define DPP4

Thank you for highlighting this. We have now defined DPP4 and included its mechanism of action (see lines 376-381).

Reviewer 2 Report

Overall the review was well-written and will be of interest for researchers within the field. 

Author Response

Overall the review was well-written and will be of interest for researchers within the field. 

We thank the reviewer for their kind words.